# Common Ground in Cooperative Communication

**Xiaoran Hao**[1,*], **Yash Jhaveri**[1,*], **& Patrick Shafto**[1,2]
[1]Department of Math and Computer Science, Rutgers University–Newark
[2]School of Mathematics, Institute for Advanced Study, Princeton

## Abstract

Cooperative communication plays a fundamental role in theories of human-human interaction–cognition, culture, development, language, etc.–as well as human-robot interaction. The core challenge in cooperative communication is the problem of common ground: having enough shared knowledge and understanding to successfully communicate. Prior models of cooperative communication, however, uniformly assume the strongest form of common ground, perfect and complete knowledge sharing, and, therefore, fail to capture the core challenge of cooperative communication. We propose a general theory of cooperative communication that is mathematically principled and explicitly defines a spectrum of common ground possibilities, going well beyond that of perfect and complete knowledge sharing, on spaces that permit arbitrary representations of data and hypotheses. Our framework is a strict generalization of prior models of cooperative communication. After considering a parametric form of common ground and viewing the data selection and hypothesis inference processes of communication as encoding and decoding, we establish a connection to variational autoencoding, a powerful model in modern machine learning. Finally, we carry out a series of empirical simulations to support and elaborate on our theoretical results.

## 1   Introduction

The cooperative communication problem is a problem among two agents; their goal is to find a pair of plans that successfully map beliefs to communicative acts and back. This problem is fundamental in two-party interaction, whether human-human, human-machine, or (expectantly) machine-machine interaction. In human-human interaction, cooperative information sharing has long been viewed as a foundation of, for example, human language (Frank and Goodman, 2012; Goodman and Stuhlmüller, 2013), cognitive development (Bonawitz et al., 2011; Jara-Ettinger et al., 2016), and cultural evolution (Tomasello, 1999). In human-machine interaction, this problem's appearance is, of course, more recent, but still its significance has been observed in, for instance, social robotics (Fisac et al., 2017), machine teaching (Zhu, 2015, 2013), reinforcement learning (Hadfield-Menell et al., 2016; Ho et al., 2016), and deep learning (Laskey et al., 2017). Researchers have proposed a number of models of cooperative communication (Shafto and Goodman, 2008; Frank et al., 2009; Shafto et al., 2012; Yang et al., 2018) and, most recently, mathematical formalizations that enable the unification and analysis of prior models (Wang et al., 2020).

The major challenge for any two agents in cooperative communication is having enough shared understanding to successfully communicate, a problem known as common ground (Clark et al., 1991). Because meaning is *inferred* from communicative acts, each agent must take into account what the other knows. So models of cooperative communication must make assumptions about the form and strength of common ground. Despite the impossibility of knowing exactly what another knows, prior models of cooperative communication uniformly assume omniscient agents (Wang et al., 2020). Ironically, this assumption with respect to common ground trivializes cooperative communication.

---

*Equal contribution.

37th Conference on Neural Information Processing Systems (NeurIPS 2023).

In this work, we propose a theory of cooperative communication that is mathematically principled and models a spectrum of common ground possibilities. This newly defined spectrum goes well beyond that of perfect and complete knowledge sharing and understanding. We formulate the cooperative communication problem under assumptions of common ground as a (family of) constrained optimization problem(s). In particular, we define different forms and strengths of common ground as different constraints on a general (family of) optimization objective(s). We draw a connection between cooperative communication and variational autoencoding notionally by viewing the data selection and hypothesis inference processes that define communication as encoding and decoding and formally through a variational approximation of our model. Our model is a strict generalization of previous models of cooperative communication, and it is set on arbitrary data and hypothesis spaces, allowing for arbitrary representations of data and hypotheses.

## 2  Theory

Communication is an ordered pair of processes between two agents, a teacher $T$ and a learner $L$, who communicate an arbitrary set of hypotheses $H$ by way of an arbitrary set of data $D$. The teacher, by definition, selects data to represent hypotheses, and the learner infers hypotheses from the teacher's selected data, all while taking into account shared knowledge, called common ground. Of course, a teacher can only represent hypotheses as well as they understand them, and, thus, a teacher naturally comes with a "prior" on hypotheses: a probability measure $\nu$ on $H$. Similarly, a learner naturally comes with a "prior" on data: a probability measure $\mu$ on $D$, as a means of representing their relative understanding of data. In preparing to communicate, therefore, each seeks a probability measure $\pi$ on a joint data-hypothesis space $D \times H$–a communication plan–that probabilistically couples data and hypotheses, formalizing how well data and hypotheses represent one another. The communication plan depends on the aforementioned priors and is subject to the teacher's and learner's assumptions about each other, part of their common ground.

**Definition 2.1** (Common Ground).  Common ground is a pair of sets of probability measures on $D \times H$, i.e., a pair of sets of communication plans: $(\mathcal{P}_T^\nu, \mathcal{P}_\mu^L) \in 2^{\Pi^\nu} \times 2^{\Pi_\mu}$, where $\Pi^\nu$ and $\Pi_\mu$ denote the sets of communication plans with hypothesis marginal $\nu$ and data marginal $\mu$.

Common ground, in turn, is a collection of all of the choices our agents have for their communication plans, accounting for their experience and understanding of data and hypotheses as well as their knowledge of each other.

We propose modeling the problem of cooperative communication under common ground, of identifying "good" communication plans, as a constrained minimization problem of a cost functional defined on pairs of communication plans:

$$\min_{(\pi_T, \pi_L) \in \mathcal{P}_T^\nu \times \mathcal{P}_\mu^L} \mathrm{CC}_{\epsilon,\delta}(\pi_T, \pi_L), \tag{2.1}$$

where, for any $\epsilon, \delta \geq 0$, the functional $\mathrm{CC}_{\epsilon,\delta} : \mathscr{P}(D \times H) \times \mathscr{P}(D \times H) \to \mathbb{R} \cup \{\pm\infty\}$ is

$$\mathrm{CC}_{\epsilon,\delta}(\pi_T, \pi_L) := \mathrm{KL}(\pi_T \,|\, \pi_L) + (\epsilon - 1)\mathrm{KL}(\pi_T \,|\, \mu_L \otimes \nu_T) + \delta \mathrm{KL}(\mu_T \,|\, \mu_L),$$

over a pair of feasible sets, $\mathcal{P}_T^\nu$ for the teacher and $\mathcal{P}_\mu^L$ for the learner. Here $\mu_i$ and $\nu_i$ denote the data marginal and hypothesis marginal of the joint probability measure $\pi_i$ respectively, for $i = T, L$. Moreover, $\mathrm{KL}(\cdot \,|\, \cdot)$ is the Kullback–Liebler divergence (or relative entropy).

By finding a pair of plans that minimize the aforementioned cost on the product of these feasible sets, our agents are cooperatively identifying plans that they can use to communicate with respect to the assumed common ground captured by the size, structure, and elements of the feasible sets. Notice that the smaller the product of the feasible sets which define their common ground, the stronger the assumptions our agents make about one another.

**Remark 2.2.**  How our agents come to determine a common ground pair is an important *independent and antecedent problem* to that which our model solves. We propose a framework which allows our agents to 1. rigorously define common ground that isn't perfect and complete, the first such rigorous definition, and 2. *with common ground established*, define and find optimal communication plans.

**Remark 2.3.**  Furthermore, given a communication plan in this generality, many possible choices of how say data might be chosen to represent a hypothesis $h$ exist. For example, if looking for a single

$d$, the mean of the conditional distribution over data given $h$ might do. This leads to an *important and subsequent problem*. What is the best way to utilize optimal communication plans to communicate once they are in hand? (Cf. Remark 3.3 below.)

If $\mathcal{P}_T^\nu \cap \mathcal{P}_\mu^L$ is non-empty, for example, our framework models situations in which agreement is possible, where agreement corresponds to the possibility that an optimal pair of communication plans consist of a single communication plan paired with itself, that is, the teacher and learner select the same communication plan in minimizing $\mathrm{CC}_{\epsilon,\delta}$. On the other hand, if $\mathcal{P}_T^\nu \cap \mathcal{P}_\mu^L$ is empty, our framework models situations in which only approximate agreement is possible.

The two parameters $\epsilon$ and $\delta$ weigh the relative importance of the three terms defining our cost functional. The first term measures disagreement between the teacher and learner; the further apart these two probability measures are, or, in other words, the more disagreement between the teacher and learner there is, the higher the communication cost. The second term, as we shall see, delineates three regimes and quantifies how uniformly the teacher might choose to represent different hypotheses as probability measures over data with respect to the learner's prior on data. The third term measures how far the teacher's induced posterior on data is from the learner's prior on data; the further apart these two probability measures on data are, the higher the communication cost.

Observe that $\mathrm{CC}_{\epsilon,\delta}(\pi_T, \pi_L)$ is increasing as a function of either $\epsilon$ or $\delta$ when all of the other terms are fixed. The first term in the definition $\mathrm{CC}_{\epsilon,\delta}$ is non-negative and vanishes if and only if $\pi_T = \pi_L$. Likewise, the third term in the definition $\mathrm{CC}_{\epsilon,\delta}$ is non-negative and vanishes if and only if $\mu_T = \mu_L$. If $\pi_T = \pi_L$, then, trivially, $\mu_T = \mu_L$. In other words, $\mathrm{CC}_{\epsilon,\delta}(\pi_T, \pi_L) \geq (\epsilon - 1)\mathrm{KL}(\pi_T \,|\, \mu_L \otimes \nu_T)$, with equality if and only if $\pi_T = \pi_L$. No matter the strength or form of common ground, we know that $\mu_L = \mu$ and $\nu_T = \nu$ for all feasible $\pi_T$ and $\pi_L$. Thus, over any pair of feasible sets,

$$\mathrm{CC}_{\epsilon,\delta}(\pi_T, \pi_L) \geq (\epsilon - 1)\mathrm{KL}(\pi_T \,|\, \mu \otimes \nu), \tag{2.2}$$

with equality if and only if $\pi_T = \pi_L$. As mentioned, therefore, we find three general situations, when considering the common ground pair $(\Pi^\nu, \Pi_\mu)$.

If $\epsilon > 1$, the coefficient $\epsilon - 1$ is positive. So taking $\mathrm{KL}(\,\cdot\,|\, \mu \otimes \nu)$ as small as possible then minimizes the right-hand side in equation (2.2). Note that $\mathrm{KL}(\,\cdot\,|\, \mu \otimes \nu)$ is non-negative and strictly convex. Thus, for all $\epsilon > 1$, $\mathrm{CC}_{\epsilon,\delta}$ over $\Pi^\nu \times \Pi_\mu$ is minimized by the unique minimizer of $\mathrm{KL}(\,\cdot\,|\, \mu \otimes \nu)$ in $\Pi_\mu^\nu := \Pi^\nu \cap \Pi_\mu$, which is $\mu \otimes \nu$, paired with itself. Indeed, for any $\pi \in \Pi_\mu^\nu$, if we take $\pi_T = \pi = \pi_L$, equality is attained in (2.2), and the right-hand side of (2.2) is equal to zero at $\mu \otimes \nu \in \Pi_\mu^\nu$. In turn, without our agents making more assumptions about one another (than that their priors are known), they will simply choose to behave uniformly with respect to the other's prior. The teacher will select data in the same way for every hypothesis: according to the learner's prior on data. Similarly, the learner will infer hypotheses in the same way for all data: according to the teacher's prior on hypotheses.

When $\epsilon < 1$, the first and third terms in the definition of $\mathrm{CC}_{\epsilon,\delta}$ compete with the second term. In particular, any admissible pair $(\pi_T, \pi_L)$ for which $\mathrm{KL}(\pi_T \,|\, \mu \otimes \nu) = +\infty$ and $\mathrm{KL}(\pi_T \,|\, \pi_L) + \delta \, \mathrm{KL}(\mu_T \,|\, \mu) < \infty$ would minimize $\mathrm{CC}_{\epsilon,\delta}$ over a common ground pair. An interpretation of this is a reflection of the importance of the teacher and learner using as much information about each other as possible beyond their priors, to maximize $\mathrm{KL}(\,\cdot\,|\, \mu \otimes \nu)$, over some version of approximate agreement, which corresponds to the finiteness of $\mathrm{KL}(\pi_T \,|\, \pi_L) + \delta \mathrm{KL}(\mu_T \,|\, \mu)$.

When $\epsilon = 1$, and taking $\pi_T = \pi = \pi_L$ for any $\pi \in \Pi_\mu^\nu$ minimizes $\mathrm{CC}_{\epsilon,\delta}$ over $\Pi^\nu \times \Pi_\mu$. Thus, in this regime, agreement is sufficient for cost minimization.

## 2.1  Common Ground via Variational Approximation

Here we consider parameterized families of probability density functions as approximations of common ground pairs, thereby producing a variational approximation of our general cooperative communication framework. This is a means of representing one manner through which our agents might attempt to disambiguate optimal communication plan pairs (by making stronger assumptions with respect to common ground, our agents may decrease the size of the set of optimal communication plan pairs) as well as resolve tractability issues (even if $\pi_T$ had probability density function $P_T$, it still may be intractable). In particular, we let $P_\theta$ denote a probability density function approximation of $\pi_T$. Similarly, we let $P_\lambda$ denote a (probability density function) approximation of $\pi_L$. The parameter $\theta$ is for the teacher and the parameter $\lambda$ is for the learner. As a byproduct, we gain a simple but

striking connection to variational autoencoding. In this scheme, the teacher's and learner's priors $\nu$ and $\mu$ must have probability density functions. They will be denoted by $g$ and $f$ respectively.

Specifically, we define variational cooperative communication under common ground as the following minimization problem: for $D, H \in \{\mathbb{R}^n, [\![n]\!] : n \in \mathbb{N}\}$ and $\epsilon, \delta \geq 0$,

$$\min_{(P_\theta, P_\lambda) \in \mathcal{P}_\theta^g \times \mathcal{P}_f^\lambda} \mathrm{L}_{\epsilon,\delta}(P_\theta, P_\lambda). \tag{2.3}$$

Here the loss functional $\mathrm{L}_{\epsilon,\delta}$ is defined on pairs of probability density functions on $D \times H$ by

$$\mathrm{L}_{\epsilon,\delta}(P_\theta, P_\lambda) := \int -P_\theta(d, h) \log f^\lambda(h \,|\, d) + \epsilon \mathrm{KL}(P_\theta \,|\, f_\lambda \otimes g_\theta) + \delta \mathrm{KL}(f_\theta \,|\, f_\lambda),$$

and common ground is given by a pair of sets of parameterized probability density functions $(\mathcal{P}_\theta^g, \mathcal{P}_f^\lambda)$; again, $f = f(d)$ and $g = g(h)$ are the learner's and teacher's prior probability density functions respectively. Also, $f_j$ and $g_j$ denote the data and hypothesis marginals of $P_j$ respectively, for $j = \theta, \lambda$. Moreover, in general, for instance, we set $f^\lambda(h \,|\, d) := P_\lambda(d, h)/f_\lambda(d)$ if $f_\lambda(d) \neq 0$ and $f^\lambda(h \,|\, d) := 0$ otherwise. Hence, $P_\lambda(d, h) = f^\lambda(h \,|\, d) f_\lambda(d)$.

As soon as a communication plan is determined by a probability density function, it admits a decomposition as the product of conditional probabilities and a prior. For example, since common ground, at minimum, requires that all of the teacher's feasible communication plans have hypothesis marginal $g$, the only variability within admissible plans comes from the induced conditional probabilities, $g^\theta(d \,|\, h) := P_\theta(d, h)/g(h)$ if $g(h) \neq 0$ and $g^\theta(d \,|\, h) := 0$ otherwise. In turn, we make the following definitions.

**Definition 2.4** (Conditional Teaching Plan). A conditional teaching plan is the family of induced conditional probabilities $\{g^\theta(d \,|\, h)\}_{h \in H}$ determined from a teacher's communication plan in $\mathcal{P}_\theta^g$.

**Definition 2.5** (Conditional Learning Plan). A conditional learning plan is the family of induced conditional probabilities $\{f^\lambda(h \,|\, d)\}_{d \in D}$ determined from a learner's communication plan in $\mathcal{P}_f^\lambda$.

While, at first glance, $\mathrm{L}_{\epsilon,\delta}$ looks to be defining a different minimization problem than $\mathrm{CC}_{\epsilon,\delta}$, even after restricting $\mathrm{CC}_{\epsilon,\delta}$ to measures defined by probability density functions, it does not.

**Theorem 2.6.** *For any $\epsilon, \delta \geq 0$, and any pair of probability density functions $(P_T, P_L)$ on $D \times H$, which determine a pair of probability measures $(\pi_T, \pi_L)$, we have that $\mathrm{CC}_{\epsilon,\delta}(\pi_T, \pi_L) = \mathrm{L}_{\epsilon,\delta}(P_T, P_L) - \mathrm{H}(g_T)$, where $g_T$ denotes the hypothesis marginal of the probability measure determined by $P_T$ and $\mathrm{H}$ denotes the Boltzman–Shannon entropy functional.*[2]

As $g_T$ represents the teacher's prior on $H$ for the communication plan $P_T$ (or $\pi_T$), which is equal to $g$ by common ground, variational cooperative communication defined by (2.3) is equivalent to cooperative communication defined by (2.1) when (2.1) is restricted to communication plans defined by probability density functions and the teacher's prior on hypotheses is denoted by $g$ (rather than $\nu$).

### 2.1.1 Cooperative Communication as Variational Autoencoding

Variational autoencoding, broadly speaking, is a probabilistic framework in which one looks to represent a space $X$ by way of a lower dimensional space $Z$. Each space is equipped with a probability density function, and the goal is to find a family of probabilistic encoders $\{q(z \,|\, x)\}_{x \in X}$, probability density functions on $Z$, one for each $x \in X$, and a family of probabilistic decoders $\{p(x \,|\, z)\}_{z \in Z}$, probability density functions on $X$, one for each $z \in Z$, such that the induced marginal on $X$ of the joint probability density function on $X \times Z$ determined by the prior on $Z$ and the probabilistic decoders is a "good" approximation of the prior on $X$.[3] These encoders and decoders are typically restricted to live in parameterized families of probability density functions, Gaussian, for example.

---

[2] All proofs can be found with the Supplementary Materials.

[3] Ideally, given a family of probabilistic decoders, the probabilistic encoders would be the $x$-induced conditional probabilities determined from the joint probability density function on $X \times Z$ formed by the product of the decoders and the prior on $Z$. These conditional probabilities are assumed to be intractable. So they need to be approximated, and the notion of "good" here has to be appropriately defined. In turn, the term variational is used slightly differently in autoencoding and cooperative communication.

Note that if we replace $X$ by $H$, $Z$ by $D$, and "represent" above by "communicate", we see that the broad idea behind variational autoencoding is exactly the broad idea behind (variational) cooperative communication.[4] Indeed, teaching is none other than encoding and learning is none other than decoding (if we consider that data are compressed representations of hypotheses), and common ground is none other than, given that the teacher's prior and learner's prior are fixed, a restriction on the admissible conditional teacher and learning plans (encoders and decoders are forced to live within some parameterized family). Moreover, the loss functional defining (2.3) is the same one used in the variational autoencoding framework by Hao and Shafto (2023) to measure the "goodness" of the approximation of the induced marginal on $X$ and the prior on $X$.

## 2.2 Previous Models

Previous models of cooperative communication assume omniscient agents and discrete data and hypothesis spaces. They define common ground, which we call *strong* common ground, as a triplet $(\mu, \nu, M)$ of two probability vectors and a matrix. These objects and the explicit manner in which they interact to determine cooperative communication will be illustrated shortly.

Here we demonstrate two things. We show strong common ground provably trivializes the problem of cooperative communication from a problem of two agents to a problem of one agent communicating with theirself. Simultaneously, we show how previous models of cooperative communication can be seen as special cases of our framework. Previous models of cooperative, from language (Frank and Goodman, 2012; Goodman and Stuhlmüller, 2013), cognitive development (Jara-Ettinger et al., 2016), robotics (Fisac et al., 2017; Ho et al., 2016), and machine learning (Zhu, 2015, 2013), were unified as approximations of entropic Optimal Transport by Wang et al. (2020). Thus, our demonstration simply involves recovering their entropic Optimal Transport framework for cooperative communication framework by appropriately constraining our model (or strengthening common ground assumptions).

Wang et al. (2020) formalize the cooperative communication problem as a pair of Boltzman–Shannon entropy regularized Optimal Transport problems on a discrete data-hypothesis space $D \times H$. More precisely, for $i = T, L$, they consider the minimization problems:

$$\min_{P_i \in \Pi_\mu^\nu} \{\langle c_i, P_i \rangle - \epsilon \mathrm{H}(P_i)\}$$

where $\epsilon \geq 0$, $c_T(d, h) := s_T(d) - \log \rho(h \,|\, d)$ and $c_L(d, h) := s_L(h) - \log \eta(d \,|\, h)$, respectively, given vectors $s_T$ and $s_L$, a non-negative $(|D| \times |H|)$-matrix $M$ with row normalization $\eta(d \,|\, h)$ and column normalization $\rho(h \,|\, d)$: $\eta(d \,|\, h)\psi(h) = M(d, h) = \rho(h \,|\, d)\varphi(d)$, for $\varphi(d) := \sum_h M(d, h)$ and $\psi(h) := \sum_d M(d, h)$, and $\mu$ and $\nu$ are probability vectors of length $|D|$ and $|H|$ respectively; $\Pi_\mu^\nu$ denotes the set of probability $(|D| \times |H|)$-matrices with column and row sums $\mu$ and $\nu$ respectively.

The unique minimizers of these two problems are the teacher's and learner's optimal communication plans respectively of the cooperative communication problem under the strong common ground triplet $(\mu, \nu, M)$. Explicitly, the matrix $M$ determines two cost matrices, one for the teacher and one for the learner through column and row normalization respectively; and the vectors $\mu$ and $\nu$ denote the learner's prior on data and the teacher's prior on hypotheses respectively.

First, note that Boltzman–Shannon entropy regularized Optimal Transport is equivalent to relative entropy (or KL-divergence) regularized Optimal Transport, with respect to the matrix $\mu \otimes \nu$ when $\mu$ and $\nu$ are the two marginal constraints, which, with a cost matrix, determine an Optimal Transport problem. So Wang et al. (2020) cooperative communication can be rewritten as, for $i = T, L$,

$$\min_{P_i \in \Pi_\mu^\nu} \{\langle c_i, P_i \rangle + \epsilon \mathrm{KL}(P_i \,|\, \mu \otimes \nu)\}, \tag{2.4}$$

with $c_i$, $\mu$, $\nu$, and $\Pi_\mu^\nu$ defined exactly as before.

Second, since cost matrices in a relative entropy regularized Optimal Transport problem can be freely "shifted" without affecting the optimal plan (Nutz, 2022), we can replace $s_T$ with any $|D|$-length vector and $s_L$ with any $|H|$-length vector. In particular, we could consider $-\log \varphi$ and $-\log \psi$ as

---

[4] In autoencoding, the space to be encoded $X$ is called the data space and the space to be decoded $Z$ is called the latent space. In cooperative communication, the space to be encoded $H$ is called the hypothesis space and the space to be decoded $D$ is called the data space. Thus, the name data is given to the opposite objects in these two worlds. We point this out explicitly to hopefully avoid any confusion that may arise from these conventions.

replacements for $s_T$ and $s_L$ respectively, and replace the cost matrices $c_i$ with the single cost matrix $-\log M$. Similarly, by adding $\log \varphi$ to $-\log M$, the cost matrix $-\log M$ defines the same entropy regularized Optimal Transport problem as the cost matrix $c(d, h) := -\log \rho(h \,|\, d)$.

**Remark 2.7.** In turn, cooperative communication under strong common ground, while formulated as two (relative) entropy regularized Optimal Transport problems, is actually just one. This is a rigorous manner in which to see that previous models of cooperative communication work under the strongest possible assumption of common ground wherein there is effectively only one agent.

In our formulation, if we take $D$ and $H$ to be discrete, $\mathcal{P}_\mu^L = \{\rho(h \,|\, d)\mu(d)\}$, $\mathcal{P}_T^\nu = \Pi_\mu^\nu$, and $\delta = 0$, we recover the a posteriori single (from the omniscient teacher's perspective) entropy regularized Optimal Transport problem that governs cooperative communication in prior work. These restrictions provide us with one manner in which to recover previous models of cooperative communication from our model. Another way is to pass to unbalanced entropy regularized Optimal Transport (Séjouré et al., 2023), by taking $\mathcal{P}_T^\nu = \Pi^\nu$ and $\delta = k \in \mathbb{N}$, and then take the limit as $k$ tends to infinity.

**Theorem 2.8.** *Let $D$ and $H$ be discrete spaces, fix $\epsilon \geq 0$, and consider the common ground pair $(\Pi^\nu, \{\rho(h \,|\, d)\mu(d)\})$ for some $(|D| \times |H|)$-row stochastic matrix $\rho$ with positive entries. Let $(P_k, \rho(h \,|\, d)\mu(d))$ be an optimal pair for (2.3) with $\delta = k \in \mathbb{N}$. Then, $P_k$ converges to $P_\infty$, and $P_\infty$ minimizes (2.4), for $i = T$ with $c_T = -\log \rho$.*

This manner of recovering previous models of cooperative communication takes advantage of the ability to decompose a communication plan into a conditional plan and a prior (cf. Definitions 2.4 and 2.5), a major practical benefit when working with the variational model Equation 2.3.

# 3    Experiments

In this section, we implement a further approximation scheme to study Equation 2.3 empirically. We sample initializations for a gradient descent based optimization scheme of $\mathrm{L}_{\epsilon,\delta}$, in PyTorch via Adam (Kingma and Ba, 2015), over $(\mathcal{P}_\theta^g, \mathcal{P}_f^\lambda)$ from a pair of probability distributions $\Theta$ and $\Lambda$ on the parameters $\theta$ and $\lambda$. We treat every parameter independently and assume that $\Theta$ and $\Lambda$ are the product of independent uniform distributions over an interval $[-p, p]$, one for each parameter. Sample initializations empirically explore common ground and the entire space of admissible communication plans. Averaging over the limiting pairs of communication plans, one for each initialization, yields an approximation to a minimizing pair of communication plans of our loss functional over the given common ground pair. In particular, we consider a multilayer perceptron-based form of common ground in which $\theta$ and $\lambda$ are neural network parameters. (We provide the full details regarding experimental setup and implementation in the Supplementary Materials.) We conduct experiments under various priors $f$ and $g$ and compare different initializations. We also analyze our model through variations on the coefficients $\epsilon$ and $\delta$. These experiments are carried out in fully discrete and semi-continuous settings. As is typical, we represent discrete spaces as collections of one hot vector.

## 3.1    The Fully Discrete Setting

Here we assume that $m = |D|$ and $n = |H|$, and we fix our common ground pair as follows: $\mathcal{P}_\theta^g = \{g^\theta(d \,|\, h)g(h)\}$ with $g^\theta(d \,|\, h) = \mathrm{Cat}(d \,|\, \alpha_\theta(h))$ and $g(h) = \mathrm{Cat}(d \,|\, \alpha)$ and $\mathcal{P}_f^\lambda = \{f^\lambda(h \,|\, d)f(d)\}$ with $f^\lambda(h \,|\, d) = \mathrm{Cat}(d \,|\, \beta_\lambda(h))$ and $f(d) = \mathrm{Cat}(d \,|\, \beta)$. As mentioned previously, $\theta$ and $\lambda$ are parameters of neural networks. For simplicity, we assume that $m = n$. So $\alpha = \beta = (1/m, \ldots, 1/m)$.

First, let us recall the **cooperative index** of two communication plans (Yang et al., 2018): $\mathrm{CI}(P_\theta, P_\lambda) := \frac{1}{n} \sum_{i,j} g^\theta(d_i \,|\, h_j) f^\lambda(h_j \,|\, d_i)$. It ranges between 0 and 1; the higher the better. The cooperative index is an established measure of the effectiveness of communication in fully discrete cooperative communication, and so, it is an important metric to consider. While we have already demonstrated that the cooperative communication as studied by Yang et al. (2018) is contained within our framework, we elaborate on this further in the Supplementary Materials.

In Figure 3.1, we study the effects of various initialization with respect to values of $p$, metrics ($\ell_1$ and cooperative index), and values of $\epsilon$. Under the common ground pair fixed in this section, we find a single unique minimizing pair for all $\epsilon > 1$ and $\delta \geq 0$. Hence, we do not consider any $\epsilon > 1$.

The two values of $p$ considered here are $0.0625$ and $0.5$. The first choice is referred to as a uniform initialization in Figure 3.1 and corresponds to the lower row; the second choice is referred to as a sparse initialization in Figure 3.1 and corresponds to the upper row.[5]

Columns (a) and (b) consider variability with respect to the size of our spaces $m$ with $\epsilon = 1$ and $\delta = 0$. With this choice of coefficient pair and our fixed common ground pair, we find that minimizing pairs are formed by a single matrix paired with itself. Hence, $\delta$ plays no role, and taking $\delta = 0$ is equivalent to taking $\delta > 0$. Column (a) plots changes in the pairwise $\ell_1$-distances between these single matrix minimizers as a function of $m$. Column (b) plots changes in the cooperative index of these single matrices paired with themselves, again, as a function of $m$. We sample each of the two distributions (uniformish and sparse) 50 times to obtain 50 initializations of the conditional teaching and learning plans. In particular, for each $m \in \{5, 10, 20, 40, 80\}$, the error bar plots the sample standard deviation of the resulting $\ell_1$-distances/cooperative indices and the central dot the mean of these resulting $\ell_1$-distances/cooperative indices, for that $m$.

Columns (c) and (d) consider variability with respect to values of $\epsilon$, which ranges between 0 and 1 with a step size of $0.1$ and $m = 20$. The two distributions (uniformish and sparse) here are sampled 100 times to obtain 100 initializations of the conditional teaching and learning plans. Here our gradient descent limits are pairs of distinct matrices, whose $\ell_1$-difference is plotted in column (c). Column (d) plots their cooperative indices. The error bars here are the standard errors of the means and the central dots are the means as determined by our 100 initializations.

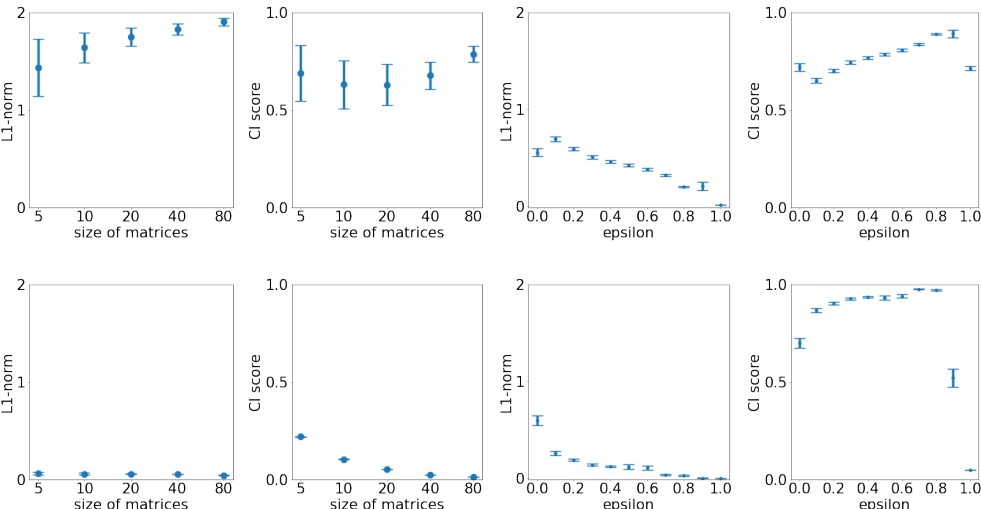

**Figure 3.1:** Columns from left to right: (a) pairwise $\ell_1$-difference with $\epsilon = 1$ as function of $m$; (b) CI with $\epsilon = 1$ as function of $m$; (c) pairwise $\ell_1$-norm as a function of $\epsilon$; (d) CI as a function of $\epsilon$. We take (a) as a verification of the existence of multiple minimizers when $\epsilon = 1$ even under rather a structured form common ground (if not the pairwise $\ell_1$-differences would all be 0). From (b), we see that sparse initializations yield higher cooperative indices. From (c) and (d), display trends of between the $\ell_1$-distances between the teacher's and learner's optimal plans as $\epsilon$ increases. Since $\epsilon = 1$ suggests a regime changes, this trend is roughly decreasing for $\ell_1$-distances and increasing for cooperative index when $\epsilon < 1$.

## 3.2 The Semi-Continuous Setting

Here we assume that $|D| = m$ and $H = \mathbb{R}^n$, and we fix our common ground pair to be $\mathcal{P}_\theta^g = \{g^\theta(d \mid h)g(h)\}$ with $g^\theta(d \mid h) = \text{Cat}(d \mid \alpha_\theta(h))$ and $g(h) = \sum_i \phi_i \text{N}(h \mid \mu_i, \Sigma_i^2)$ and $\mathcal{P}_f^\lambda = \{f^\lambda(h \mid d)f(d)\}$ with $f^\lambda(h \mid d) = \text{N}(u_\lambda(d), \sigma_\lambda^2(d))$ and $f(d) = \text{Cat}(d \mid \alpha)$; $\theta$ and $\lambda$ are parameters of neural networks. More specifically, $g$ is a mixture of $l$ Gaussians, for $l = 1, 3, 5$, with

---

[5] When $p = 0.0625$, for every $h$, $g^\theta(\cdot \mid h)$ at initialization will be close to a uniform distribution because outputs of shallow neural networks are small given small weights and biases. But if $p = 0.5$, these initial conditional probabilities will become sparse.

mean $\mu_i$ and variance $\Sigma_i^2$ sampled from the standard Gaussian, for each $i \in \{1, \ldots, l\}$. We assume that $m = 100$ and $\alpha = (0.01, 0.01, ..., 0.01)$. We begin with two definitions.

**Definition 3.1** (Reconstruction). For $h \in H$, we let $\hat{h} := \frac{1}{m} \sum_i g^\theta(d_i \,|\, h) \mu_\lambda(d_i)$, where $\mu_\lambda(d_i)$ is the mean of $f^\lambda(h \,|\, d_i)$, denote its reconstruction. The average reconstruction loss of a set of samples $\{h_i\}_{i \in I}$ is the sample mean of the $\ell_1$-difference between $h_i$ and its reconstruction: $\frac{1}{|I|} \sum_i \|h_i - \hat{h}_i\|_{\ell_1}$

**Definition 3.2** (Data Activation). A data $d$ is activated by a hypothesis $h$ if $d = \operatorname{argmax}_D g^\theta(\,\cdot\,|\,h)$. The data activation percentage is the ratio of activated data (for some $h$) over all data.[6]

**Remark 3.3.** These two definitions provide one solution to the problem raised in Remark 2.3; this solution comes from variational autoencoding.

In Figure 3.2, we restrict our attention to case where $g$ is a mixture of 3 Gaussians, $\epsilon = 1$, and $\delta = 10$. We sample 5 different initializing weights and biases for both the conditional teaching and learning plans. The value of $p$ starts at 0.01 and increases by 0.01 until 0.8. We plot changes in average reconstruction loss and activation percentage as a function of $p$. The error bars correspond to the initialization sample standard deviation and central dots correspond to the initialization sample means. As $p$ increases, standard deviation also increases, consistent with reduced common ground.

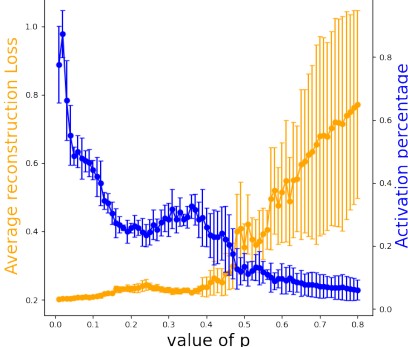

**Figure 3.2:** Variation in initialization analysis. As $p$ increases, average reconstruction loss increases and data activation percentage decreases.

In Figure 3.3, we fix $p = 0.0625$ and sample 10 initializations of our neural network parameters. We consider three cases corresponding to the number of Guassians which determine $g$. Error bars and central dots display the standard deviation and mean of these sampled initializations with respect to either average reconstruction loss or activation percentage.

Figure 3.4 plots reconstructions and data activation. The coefficient $\epsilon$ and $\delta$ are picked by the best reconstruction effectiveness, and an initialization is sampled at random with $p = 0.0625$ (cf. Figure 3.3). The size of the sample set $I$ is 6000. Column (a) plots samples in blue and their reconstructions in orange. Column (b) again plots samples in blue. The orange dots now are the means of $f^\lambda(\,\cdot\,|\,d)$s for activated data $d$; the green dots are the means of $f^\lambda(\,\cdot\,|\,d)$s for non-activated data $d$. Column (c) partitions each sampled hypothesis according to the mean associated with the data it activates. These means, colored in orange in (b), are colored in orange and outlined in black here. These plots suggest that after optimizing in $\epsilon$ and $\delta$, successful cooperative communication, defined through reconstruction coverage and continuity, is possible.

# 4 Related Work

Our framework generalizes a broad class of models that have been proposed in linguistics (Frank and Goodman, 2012), cognitive development (Jara-Ettinger et al., 2015), robotics (Hadfield-Menell et al., 2016), education (Gweon et al., 2010), cognitive science (Shafto and Goodman, 2008; Shafto et al., 2014), and machine learning (Zhu, 2015). In linguistics, the original Rational Speech Act theory papers (Frank and Goodman, 2012; Goodman and Frank, 2016) describe applications of pragmatic inference in human language. These papers are widely cited and have been applied to problems in emergent language, vision and language navigation, cultural evolution, multi-agent RL, generating and following instructions, image captioning, and machine translation. In cognitive development, Naive Utility Calculus (Jara-Ettinger et al., 2015) is also widely cited and has applications to inverse reinforcement learning and scientific models of children's behavior. In robotics, Cooperative Inverse Reinforcement Learning (Hadfield-Menell et al., 2016) and Pedagogic-Pragmatic Inference (Fisac et al., 2017) have been proposed to explain value alignment and have been applied to deep reinforcement learning for aligning with human preferences, multi-agent systems, and learning from demonstration. In education and cognitive science, the Pedagogical reasoning model explains learning

---

[6] Even if the set $\operatorname{argmax}_D g^\theta(\,\cdot\,|\,h)$ contains more than one element, PyTorch will only return one. Any plot that considers data activation suffers from this artifact. Hence, the equality.

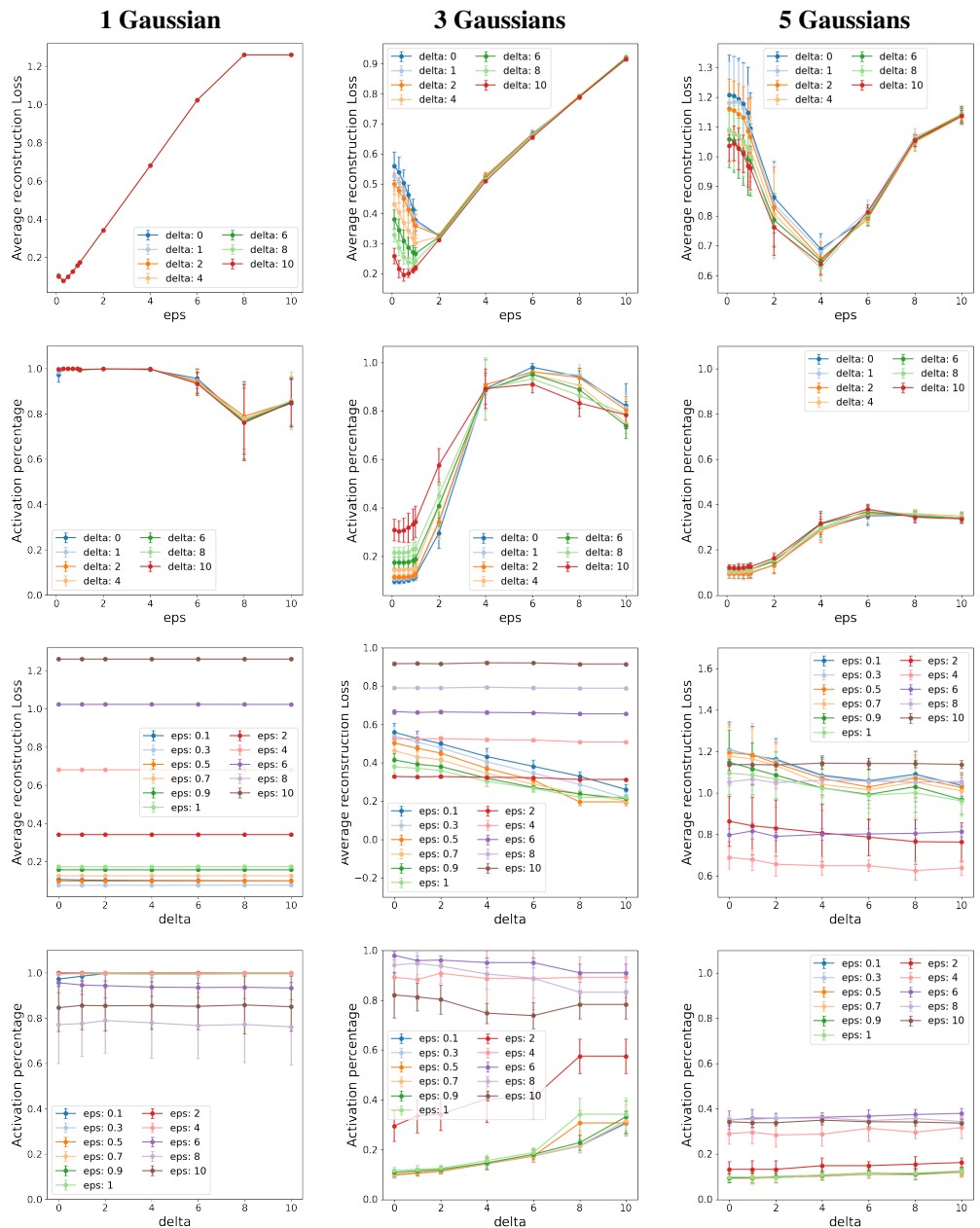

**Figure 3.3:** Rows from top to bottom: (a) V-shaped relationship between $\epsilon$ and average reconstruction loss across all $\delta$ with vertex shifting to the right as $g$ grows in complexity; (b) Near independence of data activation percentage with respect to $\delta$; (c) Stratification of average reconstruction loss by $\epsilon$ with a phase transition around $\epsilon = 1$, roughly independent of $\delta$; (d) Stratification of data activation percentage by $\epsilon$. From (c) and (d), for $\epsilon$ under a threshold that increases with the complexity of $g$, we see a roughly negative linear association between activation percentage and reconstruction loss.

from a teacher (Shafto and Goodman, 2008; Shafto et al., 2014), has been widely cited, and applied to understanding a broad array of experimental findings, informal human learning, and automated tutoring. In machine learning, the machine teaching approach is primarily a theoretical object, but has been highly influential in spawning iterative machine teaching and data poisoning approaches (Zhu, 2013, 2015). All of these models have been shown to be instances of entropic Optimal Transport (Wang et al., 2020), which our model generalizes, and so, they fail to consider imperfect common ground; using our theory, each of these works has an avenue to expand into a more realistic realm (see Section 2.2).

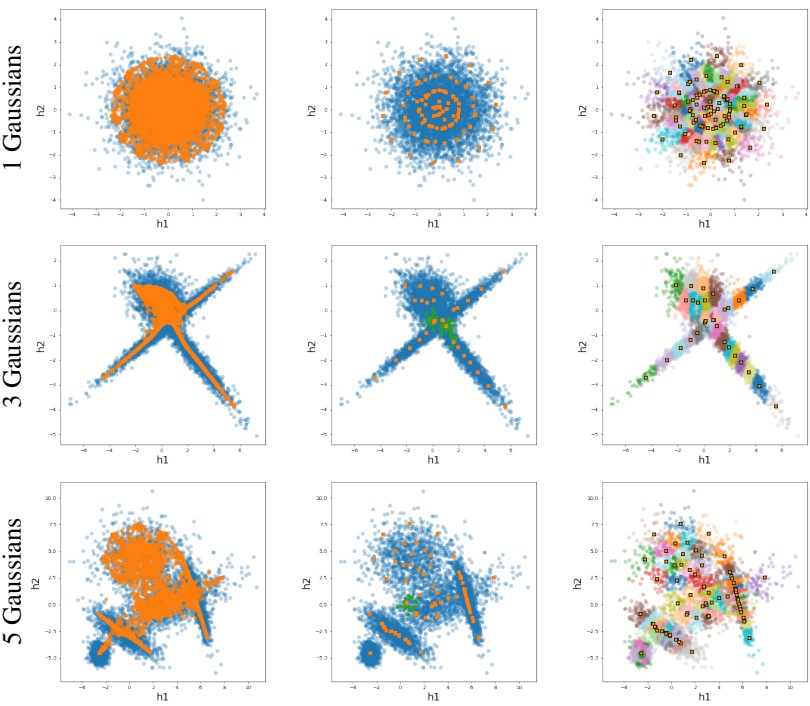

**Figure 3.4:** Columns from left to right: (a) Sample hypotheses versus reconstructions; (b) Sample hypotheses versus mean of activated data; (c) Sample hypotheses partitioned by mean of activated data. From (a), we see that hypotheses are well-reconstructed for all $g$. From (b), we see that not all data are activated. From (c), we see continuity of data selection with respect to hypotheses.

Recent proposals have used neural networks to offer an algorithmic perspective on cooperative communication. Early work by Andreas and Klein (2016) on neural pragmatics assumes, rather than derives, the encoder-decoder connection. Liu et al. (2017) formulate a teaching problem for learners who are unaware of the teacher's intent, and, thus, not cooperative, and Liu et al. (2018) assume teachers query learners to learn their beliefs. Fan et al. (2018); Wu et al. (2018) formulate teaching as a (reinforcement) learning problem. Yuan et al. (2021) consider teacher-aware learning in continuous spaces via neural networks, but do not formalize or study the common ground problem, and limit recursive reasoning between the teacher and learner to a single step.

Variational autoencoding (Kingma and Welling, 2014) is a generative probabilistic model enhanced by deep learning. Like variational autoencoding, we utilize neural networks to model conditional distributions. However, there is an important perspective shift when considering cooperative communication. Since cooperative communication is a model of two agents working together, both families of conditional probabilities are equally important, whereas in variational autoencoding the decoding process carries more emphasis, being generative.

## 5 Conclusions

We formalize common ground in cooperative communication and, thus, build a robust and accurate mathematical description of a fundamental aspect of the cooperative communication problem in two-party interaction. Our approach is an advance over prior theories which, through strong common ground, effectively assume away the most significant challenge for cooperative communication. Rooted in viewing the data selection and hypotheses inference processes that define communication as probabilistic encoding and decoding and drawing a connection between cooperative communication and variational autoencoding, this approach offers new theoretical and computational gains in the study of human-human and human-machine cooperation.

## Acknowledgments and Disclosure of Funding

This research was supported in part by DARPA awards HR0011-23-9-0050 and W912CG22C0001 and NSF grants MRI-2117429 to P.S. and DMS-1954363 to Y.J.

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
