# 6 Supplementary Materials

## 6.1 Notation and Definitions

Given a set $X$, we denote the set of probability measures on $X$ by $\mathscr{P}(X)$. While one does not require $X$ to be a measurable space (a set paired with a $\sigma$-algebra on it) in order to define $\mathscr{P}(X)$ (by considering outer measures), we assume that all the sets in this work are subsets of measurable spaces. This is non-restrictive as it is trivial to define a $\sigma$-algebra on any set. Some authors call sets paired with $\sigma$-algebras on them measure spaces, while others reserve the term measure space for triples of a set, a $\sigma$-algebra, and a measure. When $X = \mathbb{R}^n$, for some $n \in \mathbb{N}$, we assume that $\mathscr{P}(\mathbb{R}^n)$ is the set of Borel probability measures on $\mathbb{R}^n$.

When $X = D \times H$, $D$ refers to a data space, and $H$ an hypothesis space, we refer to $\mathscr{P}(D \times H)$ as the set of communication plans (on $D \times H$).

Recall, given a set $X$, the set $2^X$ denotes the power set or set of subsets of $X$.

Given two sets $D$ and $H$ and probability measures $\mu \in \mathscr{P}(D)$ and $\nu \in \mathscr{P}(H)$, we define two subsets of $\mathscr{P}(D \times H)$: the first consists of those probability measures with $D$-marginal equal to $\mu$,

$$\Pi_\mu := \{\pi \in \mathscr{P}(D \times H) : \pi(A \times H) = \mu(A) \text{ for all measurable } A \subset D\},$$

and the second consists of those probability measures with $H$-marginal equal to $\nu$,

$$\Pi^\nu := \{\pi \in \mathscr{P}(D \times H) : \pi(D \times B) = \nu(B) \text{ for all measurable } B \subset H\}.$$

Furthermore, $\mu \otimes \nu$ is the standard product measure of $\mu$ and $\nu$; it is an element of $\mathscr{P}(D \times H)$. In particular, $\mu \otimes \nu(A \times B) = \mu(A)\nu(B)$ for all measurable $A \subset D$ and $B \subset H$. In addition, we define

$$\Pi_\mu^\nu := \Pi^\nu \cap \Pi_\mu \subset \mathscr{P}(D \times H).$$

Recall the Kullback–Liebler divergence (or relative entropy) is defined as follows: given $\alpha, \beta \in \mathscr{P}(X)$,

$$\mathrm{KL}(\alpha \,|\, \beta) := \int_X \log\left(\frac{\mathrm{d}\alpha}{\mathrm{d}\beta}\right) \mathrm{d}\alpha \text{ if } \alpha \ll \beta \text{ and } := +\infty \text{ otherwise;}$$

the quotient $\frac{\mathrm{d}\alpha}{\mathrm{d}\beta}$ denotes the Radon–Nikodym derivative of $\alpha$ with respect to $\beta$.

Recall that the Boltzman-Shannon entropy functional is defined as follows: given a probability density function $f$ on $X$,

$$\mathrm{H}(f) := -\int_X f \log f.$$

## 6.2 Theory Supplement

*Proof of Theorem 2.6.* Observe that

$$
\begin{aligned}
\mathrm{L}_{\epsilon,\delta}(P_T, P_L) &= \int -P_T(d,h) \log \frac{f^L(h \,|\, d) f_L(d)}{f_L(d)} + \epsilon \mathrm{KL}(P_T \,|\, f_L \otimes g_T) + \delta \mathrm{KL}(f_T \,|\, f_L) \\
&= \int P_T \log \frac{f_L P_T}{P_L P_T} + \epsilon \mathrm{KL}(P_T \,|\, f_L \otimes g_T) + \delta \mathrm{KL}(f_T \,|\, f_L) \\
&= \mathrm{KL}(P_T \,|\, P_L) + \int P_T \log \frac{f_L}{P_T} + \epsilon \mathrm{KL}(P_T \,|\, f_L \otimes g_T) + \delta \mathrm{KL}(f_T \,|\, f_L) \\
&= \mathrm{KL}(P_T \,|\, P_L) + \int P_T \log \frac{f_L g_T}{P_T g_T} + \epsilon \mathrm{KL}(P_T \,|\, f_L \otimes g_T) + \delta \mathrm{KL}(f_T \,|\, f_L) \\
&= \mathrm{CC}_{\epsilon,\delta}(\pi_T, \pi_L) - \int \int g^T(d \,|\, h) g_T(h) \log g_T(h) \\
&= \mathrm{CC}_{\epsilon,\delta}(\pi_T, \pi_L) + \mathrm{H}(g_T).
\end{aligned}
$$

Here $g^T(d \,|\, h) := P_T(d,h)/g_T(h)$ if $g_T(h) \neq 0$ and $g^T(d \,|\, h) := 0$ otherwise. Similarly, $f^L(h \,|\, d) := P_L(d,h)/f_L(d)$ if $f_L(d) \neq 0$ and $f^L(h \,|\, d) := 0$ otherwise, where $f_L$ denotes the data marginal of $P_L$. Moreover, $f_T$ denotes the data marginal of $P_T$. The theorem follows. $\square$

*Proof of Theorem 2.8.* The set of admissible teaching plans $\Pi^\nu$ is a compact convex subset of the set of $(|D| \times |H|)$-matrices. Since $\rho$ has positive entries, $\mathrm{L}_{\epsilon,k}$ is strictly convex and bounded from below on $(\Pi^\nu, \{\rho(h\,|\,d)\mu(d)\})$. Thus, a unique matrix $P_k$ exists such that $(P_k, \rho(h\,|\,d)\mu(d))$ minimizes $\mathrm{L}_{\epsilon,k}$ on $(\Pi^\nu, \{\rho(h\,|\,d)\mu(d)\})$. For every convergent subsequence of the bounded sequence $\{P_k\}_k$, the limit is the unique $\epsilon$-entropy regularized optimal transport plan for cost $c_T$ and marginals $\mu$ and $\nu$. Hence, $P_k$ converges to $P_\infty$ the minimizer of (2.4) for $i = T$ with $c_T = -\log\rho$. $\qquad\square$

## 6.3 Experiments Supplement

### 6.3.1 Cooperative Inference via Variational Cooperative Communication

Let $H$ be a hypothesis space and $D$ be a data space, and let $\mu$ and $\nu$ denote mutually known prior distribution on $D$ and $H$ respectively. In cooperative inference, Yang et al. (2018) defines a system of two interrelated equations:

$$P_T(d\,|\,h) = \frac{P_L(h\,|\,d)\mu(d)}{P_L(h)} \text{ and } P_L(h\,|\,d) = \frac{P_T(d\,|\,h)\nu(h)}{P_T(d)},$$

where $P_L(h\,|\,d)$ is the learner's likelihood of inferring hypothesis $h$ given $d$, $P_T(d\,|\,h)$ is the teacher's likelihood of choosing data $d$ given hypothesis $h$, and $P_T(d) = \sum_{h\in H} P_T(d\,|\,h)\nu(h)$ and $P_L(h) = \sum_{d\in D} P_L(h\,|\,d)\mu(d)$ are the normalizing constants. This system is solved iteratively after initializing with a family of conditional probabilities, either $P_{L_0}(h\,|\,d)$ or $P_{T_0}(d\,|\,h)$, coming from a consistency matrix.

We claim that the two equations that govern cooperative inference is can be recovered from our model upon considering an alternating minimization scheme and when $\epsilon = 1$ and $\delta = 0$:

$$\min_{P_L\in\Pi_\mu} \min_{P_T\in\Pi^\nu} \mathrm{KL}(P_T\,|\,P_L),$$

as well as initialization.

Since the teacher's hypothesis marginal and the learner's data marginal are always fixed, our alternating minimization scheme varies conditional probabilities: the hypothesis induced family of conditional probabilities for the teacher and the data induced family of conditional probabilities for the learner. Note the other families of conditional probabilities and marginals can be found by Bayes' rule. Indeed,

$$P_T(d) = \sum_{h\in H} P_T(d\,|\,h)\nu(h) \text{ and } P_T(h\,|\,d) = \frac{P_T(d\,|\,h)\nu(h)}{P_T(d)}$$

and

$$P_L(h) = \sum_{d\in D} P_L(h\,|\,d)\mu(d) \text{ and } P_L(d\,|\,h) = \frac{P_L(h\,|\,d)\mu(d)}{P_L(h)}.$$

Therefore, assuming we start by minimizing over $P_L(h\,|\,d)$ while initializing the conditional teaching plans as follows: $P_{T_0}(d\,|\,h) = M(d\,|\,h)$, where $M(d\,|\,h)$ is the column normalization of the consistency matrix in cooperative inference, we have

$$\mathrm{KL}(P_{T_0}(h,d)\,|\,P_L(h,d)) = \mathbb{E}_{P_{T_0}(d)}[\mathrm{KL}(P_{T_0}(h\,|\,d)\,|\,P_L(h\,|\,d))] + \mathrm{KL}(P_{T_0}(d)\,|\,\mu(d))$$

And, for all $d \in D$,

$$P_{L_0}(h\,|\,d) = \operatorname*{argmin}_{P_L(h\,|\,d)} \mathrm{KL}(P_{T_0}(h,d)\,|\,P_L(h,d)) = P_{T_0}(h\,|\,d) = \frac{P_{T_0}(d\,|\,h)\nu(h)}{P_T(d)}.$$

This is the first of the two equations that define cooperative inference at step one. Similarly, we derive the second equation by fixing the initial conditional learning plans $P_{L_0}(h\,|\,d)$ and then optimizing in $P_T(d\,|\,h)$. This yields, for all $h \in H$,

$$P_{T_1}(d\,|\,h) = \operatorname*{argmin}_{P_T(d\,|\,h)} \mathrm{KL}(P_T(h,d)\,|\,P_{L_0}(h,d)) = P_{L_0}(d\,|\,h) = \frac{P_{L_0}(h\,|\,d)\mu(d)}{P_L(h)}.$$

Continuing in the alternating minimization scheme defined and outlined above for one full step until convergence, we arrive at a pair of plans $(P_T^*(d,h), P_L^*(d,h))$.

We have shown that our model and cooperative inference are theoretically equivalent under a specific choice of parameters. Now we check the equivalence empirically, assuming uniform priors on data and hypothesis and that $|H| = |D|$. In this case, communication plans are just square matrices with sum of all elements equaling to 1. The consistency matrices are randomly sampled from standard normal distribution and then normalized appropriately. The neural networks are randomly initialized. We calculate the $\ell_1$-distances between the teacher's optimal communication plans obtained from cooperative inference and those from our model. The results are listed in Table1.

**Table 1:** $\ell_1$-distances between couplings from our model and from cooperative inference.

| SIZE OF $\mathbf{M}$ | 5 | 10 | 20 | 40 | 80 |
|---|---|---|---|---|---|
| $\|P_T^* - P_{CI}^*\|_{\ell_1}$ | 3E-4± 9E-05 | 2E-4± 6E-05 | 2E-4± 5E-05 | 3E-4± 6E-05 | 6E-4± 1E-04 |

In this setting, Yang et al. (2018) shows that the optimal communication plans for the teacher and learner are the same. Translated to our framework, Yang et al. (2018) tells us that $P_T^*(d, h) = P_L^*(d, h)$. While the table suggests that our model empirically only almost recovers the (teacher's) optimal plan from cooperative inference, we believe this is a consequence of the limit of float point precision as the $\ell_1$-distance between the convergent pair $(P_T^*, P_L^*)$, which should be 0 is not 0; it is of the same size as $\ell_1$-distance between $(P_T^*, P_{CI}^*)$.

### 6.3.2 Perturbation Analysis

We investigated the stability of the alternating minimization scheme defined above as well as the model itself (joint minimization). The settings for $M$ and marginals are the same as in the previous section. We pre-train the neural network so that the conditional teaching plan is $M(d\,|\,h)$, i.e., $P_{T_0}(d\,|\,h) = M(d\,|\,h)$. We sample positive noise matrices $M_\epsilon$ by sampling each element from $\mathrm{N}(0, 1)$ and then normalize its exponential to have $\ell_1$-norm 1. We then calculate the $\ell_1$-distance between the plans if $M$ is perturbed by $s \cdot M_\epsilon$, i.e., $M_{pert} = M + s \cdot M_\epsilon$ where $s \in \{0.02, 0.04, 0.06, 0.08, 0.1\}$. We repeat the experiment 20 times for $M$ of sizes $\{5, 10, 20, 40, 80\}$. Since the pair of distributions (matrices in discrete cases) will converge to the same limiting matrix, we use $P^{opt}$ to represent the optimal plan. $P_{pert}^{opt}$ denotes the optimal plan if given $M_{pert}$.

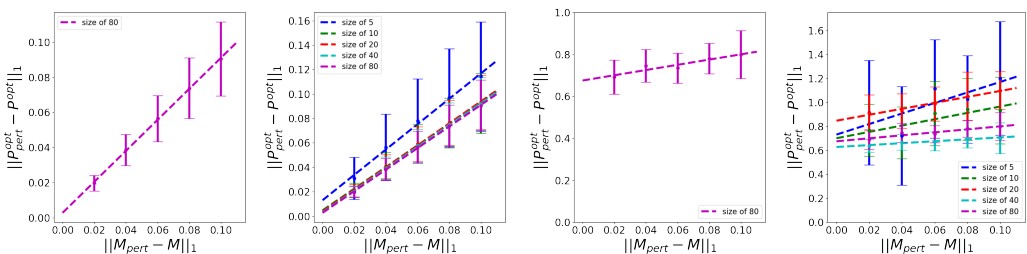

**Figure 6.1:** From left to right (a) model with alternating minimization on matrix $M$ of size 80; (b) model with alternating minimization on matrix $M$ of size $5, 10, 20, 40$, and 80; (c) model with joint minimization on matrix $M$ of size 80; (d) model with joint minimization on matrix $M$ of size $5, 10, 20, 40$, and 80. The center dots and error bars represent the mean and standard deviation over 20 experiments. The dot lines are fitted by linear regression over all points.

In Figure 6.1 (a) and (b), we see that with alternating minimization, the mean of $\|P_{pert}^{opt} - P^{opt}\|_{\ell_1}$ is approximately linear with respect to $\|M_{pert} - M\|_{\ell_1}$ with slopes around 1. And the standard deviation is increasing in $s$. The standard deviations increase as noise scales increase. This reveals the possibility of linear relationship between optimal plan and initial matrix $M$. When jointly minimization, our objective has a whole set of minimizers when considering the common ground pair $(\Pi^\nu, \Pi_\mu)$, and we attempt to approximate this by sampling many initializations. Figure 6.1 (c) and (d) suggest that, in general, our model is very sensitive to the initialization. The changes in solutions are roughly increasing over the scale of the noise.

**Table 2:** Hypothesis marginals used in experiments

| Marginals | Mean | Covariance |
|---|---|---|
| 1 Gaussian | $\mu_1 = \begin{pmatrix} 0 \\ 0 \end{pmatrix}$ | $\Sigma_1 = \begin{pmatrix} 1 & 0 \\ 0 & 1 \end{pmatrix}$ |
| 3 Gaussians | $\mu_1 = \begin{pmatrix} 1.8705 \\ -1.4432 \end{pmatrix}$ $\mu_2 = \begin{pmatrix} 0.4894 \\ -0.5546 \end{pmatrix}$ $\mu_3 = \begin{pmatrix} -0.1615 \\ 0.4268 \end{pmatrix}$ | $\Sigma_1 = \begin{pmatrix} 2.8894 & -1.8727 \\ -1.8727 & 1.2738 \end{pmatrix}$ $\Sigma_2 = \begin{pmatrix} 4.6205 & 1.9951 \\ 1.9951 & 0.8835 \end{pmatrix}$ $\Sigma_3 = \begin{pmatrix} 0.9581 & -0.1390 \\ -0.1390 & 0.2930 \end{pmatrix}$ |
| 5 Gaussians | $\mu_1 = \begin{pmatrix} -2.5486 \\ -4.5255 \end{pmatrix}$ $\mu_2 = \begin{pmatrix} 1.0328 \\ 4.2199 \end{pmatrix}$ $\mu_3 = \begin{pmatrix} -0.4467 \\ -2.6998 \end{pmatrix}$ $\mu_4 = \begin{pmatrix} 5.6779 \\ 0.3771 \end{pmatrix}$ $\mu_5 = \begin{pmatrix} 3.3328 \\ 0.1242 \end{pmatrix}$ | $\Sigma_1 = \begin{pmatrix} 0.1233 & -0.0038 \\ -00038 & 0.2399 \end{pmatrix}$ $\Sigma_2 = \begin{pmatrix} 3.1931 & -0.1261 \\ -0.1261 & 3.2483 \end{pmatrix}$ $\Sigma_3 = \begin{pmatrix} 1.4428 & -0.9623 \\ -0.9623 & 1.0269 \end{pmatrix}$ $\Sigma_4 = \begin{pmatrix} 0.2698 & -0.7867 \\ -0.7867 & 3.1732 \end{pmatrix}$ $\Sigma_5 = \begin{pmatrix} 4.5678 & 1.9727 \\ 1.9727 & 2.1902 \end{pmatrix}$ |

### 6.3.3 Experiments Details

The sample size of expectation estimation is chosen to be 600 for all semi-continuous experiments. The training iterations are fixed to 50000 for the discrete case and 20000 for the semi-continuous case. The learning rate of Adam was initially set to $5 \times 10^{-4}$ and annealed to $1 \times 10^{-5}$ gradually by cosine scheduler in PyTorch. The $\beta_1$ and $\beta_2$ parameters of Adam were set to 0.9 and 0.999 respectively.

The neural network architectures are typical variational autoencoder architectures. We have 3 fully connected layers with ReLU activation in between. The layer width is set to be 256 regardless of the depth. The output layer is stacked with a soft-max for discrete outputs and split into two sub-layers for mean and log-variance parameters of the normal distributions in semi-continuous cases.