# OpenReview forum: "Common Ground in Cooperative Communication"
_NeurIPS.cc/2023/Conference — NeurIPS 2023 spotlight_

### Official Review · Reviewer_bSj9 · 2023-07-07

**Soundness:** 3 good
**Presentation:** 3 good
**Contribution:** 3 good
**Rating:** 5
**Confidence:** 3

**Summary:**

The authors identify the problem of common ground as the core challenge in cooperative communication, where common ground means having enough shared knowledge and understanding to successfully communicate. They argue that prior models of cooperative communication uniformly assume the strongest form of common ground, perfect and complete knowledge sharing, and fail to capture the core challenge of cooperative communication.

The authors propose a general theory of cooperative communication that is mathematically principled and explicitly defines a spectrum of common ground possibilities, going well beyond that of perfect and complete knowledge sharing, on spaces that permit arbitrary representations of data and hypotheses. Also, the authors argue that their framework is a strict generalization of prior models of cooperative communication. The authors consider a parametric form of common ground and view the data selection and hypothesis inference processes of communication as encoding and decoding, and thus establish a connection to variational autoencoding. The empirical simulations support and elaborate on the theretical results.

**Strengths:**

1. Common ground is a very important concept in communication. Previous research on teacher-student communicative learning ignores this concept. I am glad to see the studies on common ground.
2. The proposed theory is quite mathematically principled. Though not very easy to follow for people without strong math background, I appreciate the authors' efforts in proposing such a theoretical framework in precise mathematical language.
3. The authors bridge the common ground and the VAE method, which is novel.
4. Results support the arguments.


**Weaknesses:**

1. Common ground is a concept from cognitive science. I cannot see any paragraphs to do some analysis on this. Is the concept used in this paper the same as the one in psychology and cognitive science? What's the same part and what's the difference? Can the proposed theory be well grounded in the cognitive theories on this concept?
2. A figure is worth a thousand words. It would help to understand the theory a lot if you could add an illustrative figure for the proposed theory. Otherwise, it's a bit hard to follow.


**Questions:**

1. Can you elaborate more on the major differences between your theory and the related works?
2. What are the potential applications of the theory?
3. What's the biggest limitation of the proposed theory?
4. Can the theory explain the underlying mechanisms for humans to form common ground?
5. When will people form common ground, and when will people recursively estimate each other's minds and cannot reach common ground? Is there a criterion to define the key moment of the emergence of common ground?

**Limitations:**

As I just mentioned, please analyze on biggest limitation of the proposed theory.

---

> ### Author Rebuttal · Authors · 2023-08-09
>
> Thank you for your detailed and comprehensive review. We have addressed questions and considerations common to all the reviewers in our global response. Here we will address the remainder of your review and considerations unique to it. If you feel something in your review has not been attended either in our global response or below, please let us know. We apologize in advance for anything missed.
>
> **Comment:** Common ground is a concept from cognitive science. I cannot see any paragraphs to do some analysis on this. Is the concept used in this paper the same as the one in psychology and cognitive science? What's the same part and what's the difference? Can the proposed theory be well grounded in the cognitive theories on this concept?
>
> **Response:** The idea of common ground originates in cognitive science, commonly traced to the paper by Herb Clark and Susan Brennan (cited in the into). Establishment and maintenance of common ground are viewed as a process in this literature, wherein speaker and listener exchange utterances in order to build common ground incrementally. In contrast with the experimental literature on perfect common ground inference, manipulations have been sequential in nature rather than in single trials. Moreover, the lack of a formalization of common ground in pragmatic models means that there not only have not, but could not, be systematic tests even in simple not scaleable models. One of our contributions is, therefore, to open the door to this possibility by formalizing, for the first time, imperfect common ground and connecting pragmatic inference with imperfect common ground to a powerful family of highly scaleable models in machine learning, variational auto encoders. With this contribution, it becomes possible to formalize the sequential reasoning process that underlies building and maintenance of common ground, which is an important direction for future work. Indeed, prior work developing the antecedent theory of cooperative inference separated the
> single (Wang, Wang, Paranamana, Shafto, 2020) from the sequential problem (Wang, Wang, Shafto, 2020) because the theoretical considerations for each are quite substantial (see also Machine teaching and Iterative
> Machine Teaching).
>
> **Comment:** Can you elaborate more on the major differences between your theory and the related works?
>
> **Response:** Previous models only consider situations when both parties have complete and perfect knowledge sharing abilities, effectively, a person communicating with themself. Furthermore, previous models, apart from one as far as we know (Wang, Wang, Paranamana, Shafto, 2020), provided algorithmic solutions to the problem of cooperative communication (with perfect knowledge sharing) without explicitly defining the problem of cooperative communication.
> We define the problem of cooperative communication and permit, for example, any solution technique that might be used to find an optimal VAE to be potentially applied to the problem of cooperative communication.
>
> **Comment:** Is there a criterion to define the key moment of the emergence of common ground?
>
> **Response:** From the point of view of our framework, yes: when the pair of sets that define common ground is fixed.

---

### Official Review · Reviewer_Xkzf · 2023-07-07

**Soundness:** 3 good
**Presentation:** 2 fair
**Contribution:** 2 fair
**Rating:** 6
**Confidence:** 3

**Summary:**

This paper models cooperative communication, particularly under imperfect knowledge sharing. The authors generalize the model of cooperative communication giving it a principled mathematical footing and elucidating the dynamics of communication by introducing insightful concepts like conditional teaching and learning plans, parametric common ground, data and hypothesis spaces, and communication cost.

The paper delves into the concept of conditional probabilities and their application in communication and learning plans. It introduces the concept of a conditional teaching plan, which is the family of induced conditional probabilities determined from a teacher's communication plan. Similarly, a conditional learning plan is the family of induced conditional probabilities determined from a learner's communication plan.

The authors discuss the concept of communication cost as a loss function, which increases as a function of either ϵ or δ when all other terms are fixed. They explain that the communication cost is non-negative and vanishes if and only if the teacher's induced posterior on data is equal to the learner's prior on data. This suggests that the communication cost is a measure of the difference between the teacher's and learner's understanding of the data. The paper also shows that their formulation of the cooperative communication problem as an encoder-decoder problem has an intuitive connection with Variational Autoencoders.

The paper also references the work of Wang et al. (2020), who formalized the cooperative communication problem as a pair of Boltzman–Shannon entropy regularized Optimal Transport problems on a discrete data-hypothesis space. This suggests that the authors are building on previous work in the field and applying it to their own research.

The authors also present experimental results, plotting reconstructions and data activation. They explain that the coefficient ϵ and δ are picked based on the best reconstruction effectiveness, and an initialization is sampled at random with p = 0.0625. The size of the sample set I is 6000. They plot samples and their reconstructions, as well as the means of f λ( · | d)s for activated and non-activated data.

**Strengths:**

Contributions and Originality: This paper offers a significant contribution to cooperative communication, particularly under imperfect knowledge sharing. The authors generalize the model of cooperative communication giving it a principled mathematical footing and elucidating the dynamics of communication by introducing insightful concepts like conditional teaching and learning plans, parametric common ground, data and hypothesis spaces, and communication cost. The paper's formulation of the cooperative communication problem as an encoder-decoder problem, and its intuitive connection with Variational Autoencoding, is a novel contribution.

Technical Soundness: The methodology is robust within the domain chosen, with a strong emphasis on thorough theoretical analysis complemented by limited practical applications. The results of the paper are convincing and well-presented. The authors provide a detailed analysis of the communication cost and demonstrate how it is influenced by various factors.

Building on Previous Work: The authors reference the work of Wang et al. and are building on previous work in the field.

Experimental Results: The authors present experimental results, plotting reconstructions and data activation. This provides empirical support to their theoretical claims.

**Weaknesses:**

Clarity and Complexity: The paper's technical complexity may limit its accessibility. More detailed justifications for the underlying assumptions may be beneficial. Although intuitive explanations are provided perhaps a more direct exposition of the complex mathematical ideas could improve its reach.

Technical Soundness: The connection between the theoretical model and actual examples of communication is not clear.  Communication doesn't happen in a vacuum, but between actual entities (humans, birds, whales, even networked computers).

Impact and Importance: While the paper presents novel concepts, it is not entirely clear how these concepts advance the state of the art or how they could be used by other researchers or practitioners. The authors could potentially improve the paper by discussing the potential applications or implications of their work in more detail.

Evaluation of Strengths and Weaknesses: The authors do not seem to provide a thorough dispassionate evaluation of the strengths and weaknesses of their work.

**Questions:**

Could you provide more details on the experimental setup, specifically on how the coefficients ϵ and δ were determined? How did you ensure that these coefficients led to the best reconstruction effectiveness? (The coefficients ϵ and δ are picked based on the best reconstruction effectiveness, and an initialization is sampled at random. However, I didn’t understand how these coefficients were determined.)

Could you elaborate on the interpretation of the experimental results? Specifically, how do the reconstructions and data activations relate to the theoretical concepts discussed in the paper? (The authors plot samples and their reconstructions, as well as the means of fλ( · | d)s for activated and non-activated data. The authors note that not all data are activated and that there is continuity of data selection with respect to hypotheses. However, I didn’t understand the interpretation of these results.)

Biggest, most abstract question: What is actually being communicated here?

**Limitations:**

The authors include a Broader Impacts section.  This does not include limitations to the current work, except for a brief mention that the current work is primarily theoretical.  They state that "while the contributions of this work are primarily theoretical, the potential positive societal implications are many," but examples of such implications are not provided.  I think it would be better to state the limitations of a purely theoretical approach to the very applied topic of communication.  What is the fidelity of the "parametric form" of common ground presented when compared to actual human studies or data-driven approaches based off the work of Clark, for example?  While the amount of math presented is impressive and I have to approach it with a presumption of correctness due to time limitations, it's unclear how this theoretical model actually describes real-world communicative phenomena.  Another way of stating this is that this paper is really about information theory rather than communication, and the connection rests on assumptions laid out by Shannon's noisy channel model, etc.

---

> ### Author Rebuttal · Authors · 2023-08-09
>
> Thank you for your detailed and comprehensive review. We have addressed questions and considerations common to all the reviewers in our global response. Here we will address the remainder of your review and considerations unique to it. If you feel something in your review has not been attended either in our global response or below, please let us know. We apologize in advance for anything missed.
>
> **Comment:** Could you provide more details on the experimental setup, specifically on how the coefficients $\epsilon$ and $\delta$ were determined? How did you ensure that these coefficients led to the best reconstruction effectiveness? (The coefficients $\epsilon$ and $\delta$ are picked based on the best reconstruction effectiveness, and an initialization is sampled at random. However, I didn’t understand how these coefficients were determined.)
>
> **Response:** The constants $\epsilon$ and $\delta$ are hyper-parameters. Each pair of $\epsilon$ and $\delta$ defines a different optimization problem and reflects different preferences with respect to the terms in the objective to be minimized. We are not searching for the best $\epsilon$ and $\delta$. Rather we try to understand how they affect minimizers as they differ. We chose a spread of $\epsilon$s and $\delta$s based on the analytical analysis of our model in the theory section, which outlines phase transitions in the model with respect to these parameters.
>
> **Comment:** Could you elaborate on the interpretation of the experimental results? Specifically, how do the reconstructions and data activations relate to the theoretical concepts discussed in the paper? (The authors plot samples and their reconstructions, as well as the means of $f_\lambda( \cdot | d)$s for activated and non-activated data. The authors note that not all data are activated and that there is continuity of data selection with respect to hypotheses. However, I didn’t understand the interpretation of these results.)
>
> **Response:** Reconstruction effectiveness and data activation are the notions/methods we defined to analyze the properties of communications plans. When we say continuity of data selection with respect to hypotheses, we mean the sampled hypotheses associated with the same data are clustered together around the mean hypothesis represented by the data.
>
> **Comment:** Biggest, most abstract question: What is actually being communicated here?
>
> **Response:** As stated in Section 2, the agents communicate general hypotheses by way of general data. Thus, the scope of communication problems encompasses virtually any problem in ML (and beyond). In the general response, we point to some literature that has used more specific formulations of our general approach. We invite you to review those papers and the citations therein.

---

> > ### Comment · Reviewer_Xkzf · 2023-08-18
> >
> > Sorry for the delayed feedback here.  Thanks for your response.  I think most of my questions are clarified and I would encourage you to include these clarifications in the final version if accepted.  I still have reservations about the overall clarity and accessibility of the paper due to the amount of math and theorems on which the main body of the paper relies (to your credit, you do address the difficulties of balancing rigor and clarity within the space constraints; I think this is something every author in this field struggles with).  I would dispute your assertion that the paper is accessible to *anyone* with an MS-level ML education.  With far more than that, it took multiple reads to fully understand how you connect the presented work to existing literature on common ground (e.g., Clark).  The mathematical formalisms in the main body are not necessarily intuitive even to someone with a long background in common ground and communication, and it's not evident in the background literature that common ground can be treated as a pure ML problem.
> >
> > Much of this might be remedied by simply reorganizing parts of the paper, and shifting some of the theorems and proofs to the appendix in favor of more narrative, example-driven, prosaic explanation in the main body.  IMO, the rigor would still be there, but the impact of the work would be more evident to someone with an MS-level background.  This is a personal preference, but I would generally say that the fewer Greek letters you use, the better job you're doing communicating your ideas (if that sounds cheeky, it is, so take from that what you will).  With the proof and theorems in the appendix, they're still there for those who care to dig deeper or formally replicate the work.
> >
> > Regarding the final question: what is actually being communicated here?—I would encourage you to include at least a few specific examples of the types and instances of communication problems you can address herein, and can restate that the scope is broadly applicable.
> >
> > I will raise my score to a 6 and I would strongly encourage you to consider such revisions if accepted to NeurIPS, or for a future submission.

---

> > > ### Author Response · Authors · 2023-08-19
> > >
> > > Thank you very much for your comments, constructive feedback, and raising of your rating. We appreciate your reservation and will most definitely, as you recommend, shift some things to the Supplement and use the newly minted space for more narrative in order to help our readers. Building a strong, stable, and inviting bridge between our abstract mathematical formalization of common ground/cooperative communication (as well as the link to ML) and our readers’ intuitive understanding or understanding coming from previous literature of these concepts is important. Furthermore, we will be sure to add some concrete, illustrative examples of what we will look to address with future work to help our readers even more.

---

### Official Review · Reviewer_UbJG · 2023-07-17

**Soundness:** 3 good
**Presentation:** 2 fair
**Contribution:** 3 good
**Rating:** 7
**Confidence:** 2

**Summary:**

> This is an emergency review for the paper. Due to the limited time, the content is shorter than a normal review, and the mathematical details are not fully checked.

This paper considers the theory of two-party cooperative communication. Compared to prior models of cooperative communication, the proposed model does not assume the strongest form of common ground and complete knowledge sharing, as a strict generalization. The problem of cooperative communication is modeled as minimizing the communication cost on the pair of plans within the admissible set. Experiments are carried out to study the properties of the model.

**Strengths:**

1. The proposed modeling of cooperative communication is a strict general form of prior models of cooperative communication. The model is shown to be compatible with previous models as a special case. It can be set on arbitrary data and hypothesis spaces and communication plans.

2. Rich experimental results on initialization, cooperative inference, perturbation, etc., with analysis are provided.

**Weaknesses:**

Lack of analysis and experiments on real scenarios (e.g. the apple experiment in [1]).
The writing in this paper sometimes needs to be clarified. It may be better to explain notations in time, and leverage subscripts to distinguish between the teacher and learner (rather than using $\theta$ and $\lambda$, $g$ and $f$). More background information and related work about cooperative communication should be provided.

[1] Wang, Pei, et al. "A mathematical theory of cooperative communication." Advances in Neural Information Processing Systems 33 (2020): 17582-17593.

**Questions:**

1. The definition of the communicative cost seems to be asymmetrical to the teacher and the learner. For example, it only considers how different the teacher chooses to represent the hypothesis with respect to the learner’s prior, but it does not consider the learner’s choice to infer. Can you provide further explanation on this asymmetry?

2. In the experiment studying initialization (Fig. 3.1), how exactly are the networks initialized to be close to a uniform distribution or become “sparse”? How are they constrained to $[-p, p]$? Can you provide more explanation or visualization to show the difference between the two choices of $p$?


**Limitations:**

The work, as well as some previous work, lacks of demonstrations or analysis of the real-world application of the theory. It is suggested to talk about or show how the theory can go beyond simple probabilistic spaces or be used in specific real-world human-robot cooperative communication cases.

---

> ### Author Rebuttal · Authors · 2023-08-09
>
> Thank you for your detailed and comprehensive review. We have addressed questions and considerations common to all the reviewers in our global response. Here we will address the remainder of your review and considerations unique to it. If you feel something in your review has not been attended either in our global response or below, please let us know. We apologize in advance for anything missed.
>
> **Comment:** The definition of the communicative cost seems to be asymmetrical to the teacher and the learner. For example, it only considers how different the teacher chooses to represent the hypothesis with respect to the learner’s prior, but it does not consider the learner’s choice to infer. Can you provide further explanation on this asymmetry?
>
> **Response:** Yes, our definition of communication cost is asymmetrical. We feel this is natural as communication itself is asymmetrical, communication is an ordered pair of processes, the teacher goes first (encodes) and the learner goes second (decodes). If one thinks of two people talking, for example, one of the two of them will speak first (encode what's in their mind into words), making them the teacher.
>
> **Comment:** In the experiment studying initialization (Fig. 3.1), how exactly are the networks initialized to be close to a uniform distribution or become “sparse”?
>
> **Response:** To be clear, we are not initializing the networks or the weights and biases of networks to be uniform or sparse. We actually try to initialize the categorical distributions parameterized by the networks. As we presented in Section 3.1, footnote 5, the output logits of shallow neural networks are near 0 if all weights and biases are initialized near 0 no matter the inputs. After applying the softmax function, the resulting distribution will be close to a uniform distribution. On the other hand, if the networks are initialized more arbitrarily, the initial logits can fall into a wide range of values. Then, after applying the softmax function to create a probability distribution, this distribution will be less uniform, i.e., a more sparse distribution.
>
> **Comment:** How are they constrained to $[-p, p]$?
>
> **Response:** We have sampled the weights and biases from a uniform distribution on interval $[-p, p]$.
>
> **Comment:** Can you provide more explanation or visualization to show the difference between the two choices of $p$?
>
> **Response:** Echoing the above response, when $p = 0.0625$, the initial categorical distribution is closer to uniform. When $p = 0.5$, the initial distribution is less uniform and, hence, more arbitrary or sparse.

---

### Official Review · Reviewer_KfB9 · 2023-07-18

**Soundness:** 2 fair
**Presentation:** 1 poor
**Contribution:** 3 good
**Rating:** 5
**Confidence:** 3

**Summary:**

The paper highlights the core challenge of cooperative communication is establishing a common ground, which refers to the shared knowledge and understanding necessary for successful communication. Existing models of cooperative communication assume perfect and complete knowledge sharing, thereby overlooking the fundamental challenge of establishing common ground. To address this limitation, the authors propose a general theory of cooperative communication that is mathematically principled and defines a spectrum of common ground possibilities, where the authors introduce a parametric form of common ground and conceptualize the communication process as encoding and decoding through data selection and hypothesis inference. To validate their theoretical results, the authors conduct a series of empirical studies. These experimental results provide additional support for the proposed framework of cooperative communication.


**Strengths:**

The paper's primary strength lies in its ability to generalize previous models of cooperative communication. By explicitly defining a spectrum of common ground possibilities, the authors propose a flexible framework that can accommodate different levels of shared knowledge, surpassing the limitations of previous setups. Additionally, the authors demonstrate clarity in explaining how their proposed method can be reduced to previous models that assume an omniscient agent with the strongest form of common ground and perfect knowledge sharing. This highlights the seamless integration of their approach with existing literature. Another strength of the paper is the discussion of the connection between their method and variational autoencoder. By drawing parallels to a powerful model in modern machine learning, the authors enhance the theoretical foundations of their framework and provide valuable insights.


**Weaknesses:**

The overall writing should be improved in the revision.

My major complain is about the writing of this paper. This paper suffers from unclear writing and excessive use of technical terms and notations without adequate definitions and explanations. This lack of clarity makes it challenging for readers to follow the paper's content and fully understand the proposed framework.

While the paper conducts two empirical experiments under two different settings (i.e., fully discrete and semi-continuous), it lacks sufficient discussion and explanation of the experimental results. The authors merely report the values without providing thorough insights into the underlying mechanisms that produce such outcomes, leaving readers with unanswered questions about the practical implications of the findings.

In addition, the figures in this paper are too small. I can barely recognize the legends and lines in Figure 3.3. Larger and more visually clear figures would improve the overall readability and understanding of the paper.

**Questions:**

See weakness.

Some notations and terms should be clearly explained and defined. For example, in line 57, why $\Pi_\mu$ and $\Pi^\nu$ use different script (i.e., one is superscript and the other is subscript)?

In the proposed framework, common ground is defined as a pair of sets of probability measures. Do these probability measures are manually defined?

What is the limitation of the proposed framework? What is the scalability of the proposed method? Does it work in a higher dimensional space comparing to the low dimensional space tested in the semi-continuous settings?

**Limitations:**

The paper lacks a systematic discussion of limitations of the proposed framework, which could benefit readers in better understanding the work. Scalability is one potential limitation. Additionally, real-world cooperative communication is more complex than communication over low-dimensional synthetic data used in the experiments, raising questions about the feasibility of deploying the proposed method in more intricate scenarios.

---

> ### Author Rebuttal · Authors · 2023-08-09
>
> Thank you for your detailed and comprehensive review. We have addressed questions and considerations common to all the reviewers in our global response. Here we will address the remainder of your review and considerations unique to it. If you feel something in your review has not been attended either in our global response or below, please let us know. We apologize in advance for anything missed.
>
> **Comment:** While the paper conducts two empirical experiments under two different settings (i.e., fully discrete and semi-continuous), it lacks sufficient discussion and explanation of the experimental results. The authors merely report the values without providing thorough insights into the underlying mechanisms that produce such outcomes, leaving readers with unanswered questions about the practical implications of the findings.
>
> **Response:** In the experiments section of the paper, we systematically vary the defining marginals of common ground sets of joint distributions as well as parameters $p$ (to explore different common ground pairs after the appropriate marginals are fixed) and pair $\delta$ and  $\epsilon$ (which shift the balance of the importance of the three terms in our cost/loss function) to describe the implications for effective communication and data activation and provide insight into the underlying mechanisms of our framework. Furthermore, in the theory section of the paper, we offer a 5 paragraph analytical discussion of the role of $\delta$ and $\epsilon$. If you have specific unanswered questions, it would be very helpful if they were articulated.
>
> **Comment:** In addition, the figures in this paper are too small. I can barely recognize the legends and lines in Figure 3.3. Larger and more visually clear figures would improve the overall readability and understanding of the paper.
>
> **Response:** We will enlarge them in the revision.
>
> **Comment:** Some notations and terms should be clearly explained and defined. For example, in line 57, why $\Pi_\mu$ and $\Pi^\nu$ use different scripts (i.e., one is superscript and the other is subscript)?
>
> **Response:** As can be read in the paper, the sets $\Pi_\mu$ and $\Pi^\nu$ are different subsets of $\mathscr{P}(D \times H)$ with respect to two conditions. Hence, the notation differs in two ways: $\mu$ versus $\nu$ and subscript verses superscript. If you see a specific piece of notation or term introduced but undefined, please let us know. That said, we will expand on our definitions in the Supplement for additional clarity.
>
> **Comment:** In the proposed framework, common ground is defined as a pair of sets of probability measures. Do these probability measures are manually defined?
>
> **Response:** Yes, our framework assumes the teacher and learner determine these sets beforehand.

---

### Author Rebuttal · Authors · 2023-08-08

We are grateful for the feedback and the effort invested in reviewing our work. We address specific considerations individually.
## Strengths (`quoted text`)
Foremost, `common ground is a very important concept`. Our work `offers a significant contribution`. Our paper's `strength lies in its ability to generalize previous models` in **3** ways. 1. `By explicitly defining a spectrum of common ground possibilities, [we] propose a flexible framework..., surpassing the limitations of previous setups` (e.g., Wang et al NeurIPS2020 (oral presentation)). We `demonstrate clarity in explaining how [our] proposed method can be reduced to previous models`, which `highlights the seamless integration of [our] approach with existing literature.` 2. Our model is `mathematically principled` and `can be set on arbitrary data and hypothesis spaces and communication plans`, which permits safe, precise, and robust study. 3. We `bridge the common ground and the VAE method, which is novel.` And `by drawing parallels to a powerful model in modern machine learning, [we] enhance the theoretical foundations of our framework and provide valuable insights.` Connecting other fields to powerful and scalable theories in ML not only offers solutions to problems therein, but also unlocks avenues for applying them in ML. Finally, we `provide empirical support to [our] theoretical claims` with `rich experimental results on initialization, cooperative inference, perturbation, etc., with analysis`.
## Considerations
**On Establishing Common Ground:** How agents decide on the common ground sets is an *independent problem*. Our framework allows agents to 1. rigorously define (imperfect) common ground and 2. *after establishing common ground*, define and find optimal communication plans. Before, this question was ill-posed: how do you know when or if common ground is reached without a concrete notion of common ground? As noted, perfect common ground effectively/arguably trivializes communication. Our paper opens the door to the systematic study of the establishment and maintenance of common ground. This text will be added to the paper.

**Technical Complexity:** The reviewers have echoed the usual tension between technical precision and readability. With new frameworks, the potential for misunderstanding tips the balance away from common exposition choices, e.g., overloading notation. Math provides a precise language for expressing ideas and eliminates ambiguity to ensure accurate communication and support logic-based rather than just intuitive conclusions. So an adequate amount of notation and technicality is necessary and inevitable. (We expect our work to be accessible to anyone with MS-level ML graduate study.) In the Supplement, we will elaborate on notation, etc.

**Application:** Our framework generalizes a broad class of models that have been proposed in linguistics (Frank et al 2012), cognitive development (Jara-Ettinger et al 2016), robotics (Hadfield-Menell et al 2016), education (Gweon et al  2011), cognitive science (Shafto et al  2008), and ML (Zhu 2015). (All cited in our intro.) In linguistics, the original Rational Speech Act theory papers (Frank et al 2012; Goodman et al 2016) describe applications of pragmatic inference in human language. These papers have 1600 citations combined and have been applied to problems in emergent language, vision and language navigation, cultural evolution, multi-agent RL, generating and following instructions, image captioning, machine translation. In cognitive development, Naive Utility Calculus has over 400 citations and applications to inverse RL and scientific models of children's behavior. In robotics, Cooperative Inverse RL and Pedagogic-Pragmatic Inference have been proposed to explain value alignment (Fisac et al  2020) and have been applied to deep RL for aligning with human preferences, multi-agent systems, and learning from demonstration. In education and cognitive science, the Pedagogical reasoning model explains learning from a teacher. These papers have been cited 1400 times and applied to understanding a broad array of experimental findings, informal human learning, and automated tutoring. In ML, the machine teaching approach is primarily a theoretical object, but has been highly influential spawning iterative machine teaching and data poisoning approaches. To our knowledge, these approaches and applications fail to consider imperfect common ground. Using our theory, each of these works has an avenue to expand into a realistic realm: incomplete or imperfect knowledge sharing. Given this vastness, with each application being a potential paper on its own, we believe that choosing one to demonstrate would detract from the broad applicability of our work. (Note, we include citation counts to illustrate the futility of tracing all of the implications and applications. The potential influence of having a principled, unambiguous, and robust definition of common ground and the problem of finding optimal communication plans is incredibly broad in ML and beyond.) We will integrate this discussion into the Related Work section of our paper to clarify the impact of our work.

**A Discussion of Limitations:** By connecting cooperative communication to VAEs, limitations that arise therein potentially arise here. While our experiments are on small-scale, low-dimensional, synthetic data, transformer-based VAEs, in the form of LLMs, are large-scale, high-dimensional, and applied on real-world data. Applying our model in more complicated scenarios amounts to being able to work with similar architectures. A question for future work is the degree to which limitations are shared or ameliorated by our approach. Another limitation relates to how we have chosen to apply gradient-based optimization schemes and leverage the power of MLP-based models to deal with constrained optimization. More comprehensive study is a question for future work. We will add this discussion to our work.

---

> ### Comment · Area_Chair_GBsv · 2023-08-19
>
> Dear reviewers,
>
> Please read all the other reviewers' discussions and the authors' feedback. Please take a moment to acknowledge the authors' rebuttal and update your rating accordingly.

---

### Author Response · Authors · 2023-08-15
**Discussion please?**

We have not received any responses from any reviewers. We would greatly appreciate discussion, as we put significant effort into addressing the reviewer's questions. Thank you!

---

> ### Author Response · Authors · 2023-08-18
> **Discussion?**
>
> Hello folks! We are most of the way through the discussion period and we haven't had any engagement. We are eager to hear from the reviewers and/or the area chairs about our responses. Thanks!

---

### Decision · Program_Chairs · 2023-09-21

**Decision:**

Accept (spotlight)

**Comment:**

The paper presents a general theory of cooperative communication that addresses the challenge of establishing common ground, which is crucial for successful communication. The reviewers' assessments highlight the theoretical and conceptual contributions of the paper, while also pointing out areas where improvements could be made.

The proposed theory's ability to generalize previous models of cooperative communication is recognized as a major strength. The introduction of a parametric form of common ground and the exploration of its connection to variational autoencoders are considered novel and contribute to the paper's originality.

However, the reviewers also highlight some areas for improvement. The clarity of the writing is noted as an issue, with concerns about excessive technical terms and notations without adequate explanations. The presentation could be improved by using more illustrative figures to help readers follow the proposed theory.

Additionally, the paper is encouraged to provide more context by discussing how the concept of common ground in the proposed work relates to its cognitive science and psychology counterparts. The limitations and potential applications of the theory should be discussed in more detail, and the connection between the theoretical model and real-world communication scenarios could be clarified.

Overall, the reviewers acknowledge the theoretical contribution of the paper but also emphasize the need for improvements in terms of clarity, illustration, and contextualization. With revisions addressing these concerns, the paper could be accepted as a spotlight paper.